# The Role of Retinal Ganglion Cell Structure and Function in Glaucoma

**DOI:** 10.3390/cells12242797

**Published:** 2023-12-08

**Authors:** Kathy Ming Feng, Ta-Hsin Tsung, Yi-Hao Chen, Da-Wen Lu

**Affiliations:** Department of Ophthalmology, Tri-Service General Hospital, National Defense Medical Center, Taipei 11490, Taiwan; fengk57@gmail.com (K.M.F.); happyshino5@gmail.com (T.-H.T.); keanechen18@gmail.com (Y.-H.C.)

**Keywords:** retinal ganglion cells, glaucoma, neuroprotection, neurodegeneration, optical coherence tomography, imaging

## Abstract

Glaucoma, a leading cause of irreversible blindness globally, primarily affects retinal ganglion cells (RGCs). This review dives into the anatomy of RGC subtypes, covering the different underlying theoretical mechanisms that lead to RGC susceptibility in glaucoma, including mechanical, vascular, excitotoxicity, and neurotrophic factor deficiency, as well as oxidative stress and inflammation. Furthermore, we examined numerous imaging methods and functional assessments to gain insight into RGC health. Finally, we investigated the current possible neuroprotective targets for RGCs that could help with future glaucoma research and management.

## 1. Introduction

Glaucoma is a multifactorial eye disease defined by the progressive degeneration of retinal ganglion cells (RGCs) and their axons, eventually leading to irreversible vision loss. It has been projected that the worldwide prevalence of individuals affected by glaucoma will experience a substantial growth of 74% between the years 2013 and 2040 [1]. Glaucoma poses significant clinical and public health challenges as one of the leading causes of blindness around the globe. While elevated intraocular pressure (IOP) remains a significant risk factor and therapeutic target for glaucoma, it is becoming increasingly apparent that other factors may play a role in the disease’s pathogenesis and progression. The structure and function of RGCs, which serve as the ultimate output neurons of the retina and transmit visual information to the brain, are among the most crucial components being studied.

The study of RGCs provides distinct insights not only into the mechanistic foundations of glaucoma but also into potential avenues for intervention and treatment. RGC apoptosis and axonal loss within the inner retina may implicate the earliest manifestation of glaucoma and exhibits a direct correlation with the clinical severity of the disease [2,3]. Apoptosis plays an essential role in maintaining homeostasis during normal development and aging; however, pathologic apoptosis is associated with age-related macular degeneration, retinitis pigmentosa, and neurodegeneration [4,5]. RGC damage can also be ascribed to various contributing mechanisms, including elevated IOP, impaired mitochondrial activity, oxidative stress, activation of glial cells, and excitotoxicity. With advances in imaging technology and functional assessment instruments such as optical coherence tomography, confocal scanning laser ophthalmoscopy, and adaptive optics, it is now possible to evaluate RGCs in greater detail [2]. A comprehensive understanding of how RGCs are affected by glaucoma could potentially transform the disease’s diagnosis and management, allowing ophthalmologists to provide more precise and personalized care to patients.

This review examines the structure and function of RGCs in the context of glaucoma. This includes topics ranging from the anatomy and physiology of RGCs to their susceptibility to glaucomatous conditions. In addition, this review investigates the most recent diagnostic imaging techniques and functional assessment methods. The review concludes by examining emerging therapeutic approaches that target RGCs, focusing on their clinical implications and future potential.

## 2. Basic Anatomy and Function of Retinal Ganglion Cells

RGCs play a crucial role in transmitting visual stimuli from the retina to the brain. From a morphological standpoint, RGCs consist of a cellular body, a complex dendritic structure, and a single axon. The primary function of dendrites is to receive synaptic inputs from bipolar and amacrine cells, which are intermediary neurons responsible for acquiring signals generated by photoreceptors [5,6,7] (Figure 1). The presence of hierarchical circuitry allows RGCs to engage in intricate signal integration and processing, leading to the conversion of photoreceptor responses into action potentials. The action potentials propagate through the RGC axons, which together constitute the optic nerve. They travel through the optic disc and optic canal to establish synaptic connections in the lateral geniculate nucleus (LGN), located in the thalamus [6]. The visual information is transmitted from the LGN to the primary visual cortex (V1) for the purpose of undergoing higher-level processing and integration. Therefore, RGCs play a crucial role in connecting the initial light detection in the retina to the following brain visual perception.

In the human retina, a variety of RGCs are present, each specializing in distinct visual functions. The presence of heterogeneity among RGCs is a fundamental quality, as these cells can be differentiated into multiple subtypes based on their distinct morphological and physiological characteristics. Midget RGCs (P-Cell), comprising approximately 80% of RGCs, are primarily associated with the processing of visual information at a high level of detail and the discriminating of colors [8]. Parasol RGCs (M-Cell), distinct from midget cells, are known to play a role in the detection of motion and changes in luminance [9]. Midget RGCs receive inputs primarily from single cone photoreceptors and exhibit a sustained response to stimuli; the axons of midget RGCs project to dorsal parvocellular layers of the LGN. Parasol RGCs exhibit a larger dendritic field compared to midget RGCs and have a transient response to visual stimuli; the axons of parasol RGCs project to the ventral magnocellular layers of the RGC [10]. Small bistratified RGCs, which constitute about 5–8% of the total, are an additional prominent subtype that receives synaptic inputs from both ON and OFF bipolar cells, allowing them to integrate signals from both pathways and contribute to the encoding of color contrast, particularly along the blue–yellow axis. These cells are components of the koniocellular pathway, implicated in color processing and integrating diverse visual features [8,11]. Smooth monostratified RGCs, divided into ON and OFF cells, can relay spatial information in a complex manner [12]. Melanopsin-containing intrinsically photosensitive RGCs (ipRGCs), essential for non-visual functions, represent about 1% of the total RGC population [13]. Additionally, there are miscellaneous RGCs that defy existing classifications, underscoring the retina’s intricate complexity [14]. Each specific subtype of RGCs makes distinct contributions to many areas of visual perception, encompassing spatial resolution, contrast sensitivity, and chromatic discrimination [15,16]. Furthermore, it is important to note that certain subtypes of RGCs exhibit varying levels of vulnerability to degenerative alterations in glaucomatous circumstances. For example, in a pressure-induced environment, midget and parasol cells undergo degeneration that begins with dendritic arbor and concludes with cell soma shrinkage [10]. As a result, it is crucial to develop a comprehensive understanding of the distinct functions and contributions of these subtypes.

## 3. Pathophysiology of Glaucoma and Current Theories on Retinal Ganglion Cell Vulnerability and Functionality in Glaucoma

Glaucoma is a heterogeneous group of ocular disorders that culminates in RGC degeneration and visual field loss. The mechanism underlying glaucomatous optic neuropathy is multifaceted, with theories on RGC vulnerability that include mechanical, vascular, excitatory, neurotrophic factor deprivation, and immune and inflammation mechanisms (Table 1) [17].

### 3.1. Mechanical Theories

The pathophysiology of glaucoma is complex, involving both mechanical and vascular components. Mechanical strain and compressive forces are exerted at the lamina cribrosa, recognized as the most susceptible region of a pressurized eye, subsequently affecting the structural stability and functionality of RGC axons [18]. Research showed that elevated IOP from 5 to 50 mmHg over 24 h resulted in an average posterior deformation of the central lamina measuring 79 μm in human donor eyes and posterior displacement of the central lamina of 12 μm in response to an acute increase of IOP from 10 to 25 mmHg in human eyes [19,20]. In addition, connective tissue of the peripapillary sclera and scleral canal wall are also responsible for bearing the forces generated by the IOP. The structural stiffness of the sclera plays a significant role in determining the deformation of the lamina cribrosa. Compliant sclera enables expansion of the scleral canal in response to elevated IOP, which results in tightening of the laminar beams within the canal and increases lamina resistance against posterior deformation. On the other hand, a rigid sclera exhibits limited capacity for canal expansion, hence necessitating the sole reliance on the structural rigidity of the lamina cribrosa to withstand the stress associated with elevated IOP [20]. Therefore, the sclera and lamina cribrosa serve as the load-bearing tissues of the optic nerve head [21].

### 3.2. Vascular Theories

The aforementioned discussion on mechanical theory suggests axonal compression at the lamina cribrosa, hindrance of axoplasmic flow, and disruption of retrograde neurotrophin transport to RGCs, ultimately resulting in cellular demise. Another aspect of RGC vulnerability is vascular insufficiency, where it is hypothesized that impaired perfusion pressure, disruption of vascular autoregulation, and loss of neurovascular coupling can lead to RGC dysfunction and death. In a study conducted by Shiga et al., the results of laser flow speckle flowgraphy indicated a statistically significant decrease in ocular blood flow in patients with preperimetric glaucoma compared to normal subjects [22]. 

Ocular perfusion pressure (OPP) is defined as the sum of the systolic arterial blood pressure and one third of the difference between the systolic and diastolic pressures, minus the IOP. OPP signifies the oxygen supply and blood flow to the optic nerve; thus, it has been hypothesized that a reduction in OPP could increase the optic disc’s susceptibility to damage, thereby increasing the likelihood of glaucoma progression or development [23]. Many studies have revealed a positive correlation between a reduction in OPP and an elevated prevalence of open-angle glaucoma [24,25,26]. Additionally, previous research utilizing microspheres indicated a specific reduction of volume flow within the prelaminar and anterior laminar capillary beds when the OPP fell below 30 mm Hg [27,28]. Regulation of ocular blood flow is affected by OPP, temperature, and neural function, and blood flow to the optic nerve head is mainly regulated by endothelial cells and circulating hormones [29]. 

Vascular autoregulation refers to the regulatory mechanism that counteracts fluctuations in perfusion pressure, functioning optimally within a specific range of perfusion pressure. This mechanism can be compromised in conditions like glaucoma as well as other diseases, including diabetes mellitus. Various factors are involved in autoregulation, such as metabolic, myogenic, and neurogenic components. Direct autoregulation of ocular blood flow is difficult to assess, and many techniques have been used to measure ocular circulation, such as color Doppler imaging, scanning laser Doppler flowmeters, and optical coherence tomography (OCT). A study revealed individuals with glaucoma exhibited inadequate compensatory reaction of flow velocities in the short posterior ciliary artery (SPCA) to alterations in body position compared with healthy individuals, suggesting inadequate autoregulatory control may serve as a contributing factor to glaucoma pathogenesis [30]. 

The central nervous system has a strong correlation between neuronal activity and blood flow, indicating a high level of coordination. Neurovascular coupling is the phenomenon wherein heightened neural activity leads to a concomitant augmentation in blood flow to the appropriate region [31]. A study demonstrated a lower cerebral blood flow and functional connectivity strength coupling in glaucoma patients compared to controls, and the reduced ratio was significantly correlated with visual field defects and glaucoma stage, suggesting impaired neurovascular coupling in glaucoma patients [32].

Numerous mediators orchestrate the regulation of ocular blood flow in glaucoma, which is a critical determinant of the health of the RGC and optic nerve head. Central among these mediators is endothelin-1 (ET-1), a potent vasoconstrictive peptide, which is elevated in aqueous and plasma concentrations of glaucoma patients [33,34,35,36,37]. Clinical studies revealed worsening visual fields have higher plasma ET-1 compared to normal visual fields [35], and animal studies demonstrated that low doses of ET-1 can induce glaucomatous changes in primates [38]. Increased ET-1 levels result in an increase in IOP, which decreases ocular blood flow and astrocyte proliferation; consequently, this may ultimately lead to the degeneration of RGCs [39]. A recent study utilized advanced imaging techniques, specifically OCT and OCT angiography, to assess vessel density in the peripapillary and macular regions of patients with glaucoma, revealing an inverse relationship with systemic ET-1 concentrations [37]. Multivariate analyses indicated that IOP plays a less predictive role in the diminishment of retinal blood flow when compared to ET-1 levels in a glaucoma context. Elevated peripheral ET-1 has been identified as a potential risk indicator for monitoring vascular alterations in the optic nerve head of eyes affected by glaucoma. Conversely, nitric oxide acts as a pivotal vasodilator by stimulating cyclic guanine monophosphate (cGMP) to reduce intracellular calcium and thereby relaxation of smooth muscle cells and pericytes [40]. In glaucoma patients, nitric oxide availability is reduced, which can cause a disruption in the equilibrium between vasoconstriction and vasodilation and ultimately results in a reduction of blood flow to the optic nerve head [41].

### 3.3. Excitotoxicity Theory

Research into the role of excitotoxicity, which is mediated by excessive glutamate release and subsequent overstimulation of N-methyl-d-aspartate (NMDA) receptors on RGCs, has revealed the molecular and cellular pathways that contribute to RGC injury in glaucoma. Glutamate serves as a primary neurotransmitter inside the retinal network, facilitating communication between the photoreceptors, bipolar cells, and RGCs [42]. The process of glutamate neurotransmission is intricately regulated by inhibitory and modulatory neurotransmitters, including γ-aminobutyric acid (GABA), glycine, acetylcholine, dopamine, serine, substance P, and several neuropeptides [43]. Excessive glutamate release causes overactivation of the NMDA receptor, which induces calcium influx, oxidative stress, and mitochondrial dysfunction, all of which contribute to the degeneration of RGCs. An increase in glutamate has not been consistently seen in retinal pathologies such as glaucoma [44]; however, due to the progressive nature of glaucoma, glutamate levels may not significantly increase acutely or accumulate during the course of the disease and may only rise in specific parts of the retina or optic nerve.

Generally, when glutamate is released, the initial depolarization of the cell membrane through α-amino-3-hydroxy-5-methylisoxazole-4-propionate (AMPA) receptor activation leads to an elevation in the likelihood of NMDA receptor opening. This triggers a transient influx of calcium ions into the intracellular space. Calcium can also be released via the inositol 1,4,5-trisphosphate receptors (IP3R) and the ryanodine receptors (RyR) that are present on the membrane of the endoplasmic reticulum. Functional mitochondrial metabolism plays a crucial role in maintaining intracellular calcium homeostasis through the reuptake of calcium by the Sarco-Endoplasmic Reticulum Calcium ATPase (SERCA) and the extrusion of calcium to the extracellular space via the sodium-calcium ATPase. Furthermore, the extrusion of calcium is facilitated by the sodium-calcium exchanger (NCX) through the utilization of the physiological sodium gradient. The failure of RGCs to adequately regulate or remove excessive calcium ions disrupts the homeostatic mechanisms, which ultimately culminates in cellular demise. The involvement of NCX in the process of retinal cell death triggered by NMDA and ischemia-reperfusion has been suggested [45,46]. High levels of intracellular calcium along with heightened activity of catabolic enzymes can trigger a cascade of events, leading to RGC apoptosis and necrosis [47,48]. The subsequent cascade encompasses mitochondrial membrane depolarization, activation of caspases, generation of harmful oxygen and nitrogen free radicals, and manifestation of cellular toxicity.

Normal glutamate excitotoxicity is minimized by its quick absorption by glutamate transporters in glial cells surrounding RGCs. The majority of glutamate transporters in the retina are located at the synaptic cleft in order to quickly remove released glutamate and prevent overflow [49]. Glutamate uptake by Müller cells via glutamate/aspartate transporters (GLAST or EAAT1) is crucial for maintaining physiological glutamate levels [50]. Retinal glia and neurons produce several glutamate transporters, such as GLT-1 (EAAT2), excitatory amino acid carrier 1 (EAAC1 in rats or EAAT3 in humans), EAAT4, and EAAT5 in rats. Decreased retinal GLAST has been correlated with glaucoma in rats and mice [51,52], and GLAST-deficient mice displayed loss of RGCs [53]. 

The profound vulnerability of RGCs is underscored by the imbalance between neurotransmitter release and uptake, as well as the cascade of intracellular events triggered by excessive receptor stimulation.

### 3.4. Neurotrophic Factor Deprivation Theory

According to the neurotrophic factor deprivation theory, the interruption of axonal transport results in a diminished supply of neurotrophic factors, which are essential for the survival of RGCs. Neurotrophins, which are diffusible trophic molecules, enhance the survival of mature neurons in the central nervous system (CNS), which degenerate in response to an extensive range of stimuli. Brain-derived neurotrophic factor (BDNF) has garnered significant interest within the field of neurotrophins due to its influential impact on the survival of RGCs. The initial evidence of BDNF’s neuroprotective properties in RGCs occurred in 1986 [54]. BDNF is produced locally in the retina by RGCs and glia, but it is also strongly expressed in the superior colliculus, where it is retrogradely transported to the optic nerve head and to the cell bodies of RGCs [49,55]. The effects of BDNF are mediated through tropomyosin receptor kinase B (TrkB) on the presynaptic cell surface, and TrkB activation leads to three major signaling cascades (mitogen-activated protein kinase (MAPK), phosphatidylinositol 3-kinase (PI3K), and phospholipase Cγ (PLC-γ)), which are responsible for promoting the survival of RGCs [56]. Elevated IOP and the resulting mechanical strain on the optic nerve head may hinder the retrograde transport of BDNF in glaucoma. When the stressed RGC is deprived of BDNF, Jun-N terminal kinase (JNK)-mediated signaling may be triggered, which may activate the proapoptotic BCL-2 family of proteins and cause mitochondrial dysfunction [55,57,58]. Numerous studies have demonstrated a correlation between RGC loss and impaired BDNF-TrkB signaling [59,60,61]. Consistent clinically, BDNF deficit was demonstrated in serum, aqueous humor, and lacrimal tears of patients with early glaucomatous change [62,63]. Apart from BDNF, additional neurotrophic factors have been linked to RGC health, including ciliary neurotrophic factor (CNTF), nerve growth factor (NGF), and glial cell line–derived neurotrophic factor (GDNF) [57].

The multifaceted nature of neurotrophic factors in the retina emphasizes the importance of its support in RGC health. 

### 3.5. Oxidative Stress and Inflammation

Oxidative stress and neuroinflammation are implicated in the pathogenesis of glaucoma, where reactive oxygen species (ROS) and inflammatory mediators precipitate cellular damage and contribute to an excitotoxic environment, thereby exacerbating RGC death. Multiple factors can contribute to increased ROS production in glaucomatous eyes. Ischemia-reperfusion injury, elevated IOP, and mitochondrial dysfunction are among the principal factors that contribute to the overproduction of ROS in the optic nerve head and retina [49,64]. The principal roles of mitochondria encompass the synthesis of adenosine triphosphate (ATP) via the oxidative phosphorylation pathway and the orchestration of apoptotic cell death [65]. ROS are consistently generated within mitochondria via the electron transport chain. However, enzymatic degradation of neurotransmitters, neuroinflammatory mediators, and redox reactions can also contribute to ROS production [66]. In the anterior chamber, the trabecular meshwork (TM) is the most susceptible tissue to oxidative injury [67]. Oxidative stress on the TM can induce numerous detrimental effects, including altering cell-cycle progression [68,69], changes in the extracellular matrix [70], rearrangement of TM cell cytoskeletal structures, loss of cell-matrix adhesion [71], and alteration of membrane permeability [72]. Additionally, it can induce inflammatory cytokine release and initiate RGC apoptosis [49]. Mitochondrial dysfunction augments the production of ROS while concurrently diminishing the synthesis of ATP. Neuronal cells, which demand substantial energy to sustain electrochemical gradients crucial for signal conduction, are exceedingly susceptible to perturbations in mitochondrial function. The generation of ATP predominantly occurs within the mitochondria via the oxidative phosphorylation mechanism of the electron transport chain, complemented by the glycolytic process [73]. Defects in mitochondrial complex 1 were found to induce ROS release and decrease ATP levels in human glaucomatous TM cells [74]; a study found that RGC death occurred 3–4 months after elevated IOP, with an 18% reduction in mitochondrial membrane potential [75]. Dynamin-related protein 1 (Drp1), which is present on the outer mitochondrial membrane, has been shown to play a role in glaucoma [76]. A recent study showed that the ERK1/2-Drp1-ROS axis can cause mitochondrial dysfunction and apoptosis in RGCs [77]. The involvement of mitochondria in glial neuroinflammation processes is facilitated by the activation of NF-κB through the generation of mitochondrial ROS, resulting in the production of inflammatory cytokines [78].

In response to cellular injury and oxidative stress, numerous pro-inflammatory mediators are upregulated. Normally, the eye is regarded as an immune-privileged site; however, the permeability of the blood–retina barrier may be altered by factors such as oxidative stress, vascular endothelial growth factor, and inflammation [79]. In addition to chemokines and adhesion molecules, cytokines such as interleukins 1β (IL-1β) and tumor necrosis factor-alpha (TNF-α) exhibit increased levels in glaucomatous tissues [80]. Elevated IOP can trigger an innate immune response with researchers finding an increase in microglia activity and cell density as well as expression of the complement of C1q in the retina and optic nerve in mice with glaucoma [81,82]. Adaptive immunity of glaucoma was seen when heat shock protein (HSP)-specific memory T cells were induced by commensal microflora [83]. HSP-27, HSP-60, and HSP-70 autoantibodies have been found in the sera of glaucoma patients [84]. In addition, microglial cells are activated in response to elevated IOP by producing cytokines, mediators, and enzymes that can alter the ECM [85]. Microglial cells can exhibit two phenotypes, M1 (pro-inflammatory) or M2 (neuroprotective); however, in glaucomatous eyes, the majority of microglia do not show the M2 phenotype, and the activities of microglia may have detrimental consequences for the glaucomatous optic disc [86]. 

In conclusion, the RGC vulnerability hypotheses in glaucoma are multifaceted, encompassing mechanical, vascular, metabolic, excitotoxic, and inflammatory dimensions. The combination of these theories highlights the complexity of glaucomatous neurodegeneration and necessitates a holistic understanding of the changes in RGC functionality, thereby guiding the development of nuanced therapeutic strategies aimed at reducing RGC vulnerability and preserving visual function.

## 4. Imaging Technique for Assessing Retinal Ganglion Cells

### 4.1. Structural Imaging Modalities

The elucidation of RGC structure and function in glaucoma requires cutting-edge imaging techniques that provide precise, dependable, and non-invasive evaluations. OCT is a pioneering modality introduced in the early 1990s, as it allows for the detailed visualization of retinal layers and quantification of the retinal nerve fiber layer (RNFL) [87]. It first produced images through measurement of the echo time delay of light that was backscattered and reflected from ocular tissues [88]. However, due to its suboptimal resolution and slow scan acquisition, visualization of RGCs was difficult. Later, introduction of spectral-domain OCT (SD-OCT), also known as Fourier-domain OCT (FD-OCT), allowed for higher resolution, faster acquisition scan rate, and enhanced reproducibility. Another type of FD-OCT, swept-source OCT (SS-OCT), permits scanning velocities up to 200,000 A-scans/s and employs a laser that rapidly transverses a wide range of frequencies. A photodetector is utilized to capture the interference pattern instead of the spectrometer used in SD-OCT [89]. The capability of OCT to detect subtle alterations in RNFL thickness provides invaluable insights into early glaucomatous changes and is a crucial tool for monitoring disease progression and treatment response [90]. 

Moreover, confocal scanning laser ophthalmoscopy (cSLO) is widely used to evaluate the optic nerve head topography, producing detailed images of the optic disc and its adjacent tissues that are crucial for assessing structural changes indicative of glaucomatous damage [91]. It consists of a 670 nm diode laser, which uses a narrow laser beam and small aperture to limit scattered light from outside the focal plane [2]. Unlike standard fundus imaging, this provides excellent lateral resolution for topography images but poor axial resolution. 

Adaptive optics (AO) compensates for wavefront distortions with a deformable mirror and reduces the optical aberrations that can cause blurring and image artifacts in ophthalmic imaging [92]. In 1997, AO was integrated with a fundus camera, with scanning laser ophthalmoscopy (SLO) in 2002, and with OCT in 2005. Pairing AO with OCT enables three-dimensional, diffraction-limited images of the optic nerve head, lamina cribrosa, and retina. It is noteworthy that individual nerve fiber bundles can be distinguished in these images, even after repeated sessions, which renders this method appealing for investigating the progression of glaucoma [93]. 

### 4.2. Functional Imaging Modalities

Functional evaluation of RGCs is of equal importance, and developments in electroretinography (ERG) have enabled the isolated evaluation of RGC function, providing a unified view of both structural and functional integrity. Recent findings have shown that pattern reversal ERG (PERG), photopic negative response (PhNR), and multifocal ERG can be objective measures of RGCs [94]. At present, PERG is one of the most recognized ERG methods for the identification of glaucoma. Earlier investigations have suggested that it could be particularly advantageous when used in conjunction with the diagnosis and treatment of individuals suspected of having glaucoma, particularly those with normal or borderline visual fields and/or RNFL thickness [95,96]. A study found that it takes 2 years for PERG to change by 10% and 10 years for RNFL to change by 10%, indicating a time lag of 8 years between changes in PERG and RNFL in glaucoma suspects [95]. Another study identified individuals with glaucoma four years prior to the onset of visual field alterations with a sensitivity and specificity of 75% and 76%, respectively [96]. Furthermore, the prognostic capability of PERG in the evaluation of glaucomatous changes has been observed [97,98]. Research indicated that lower pERG N95 amplitudes correlate with retinal nerve fiber layer defects in preperimetric glaucoma and decrease progressively from normal controls to perimetric glaucoma patients, suggesting that pERG N95 amplitude could be a useful metric for early glaucoma detection [97].

PhNR is a slow negative wave after the b-wave in ERG under photopic conditions that originates from RGCs [99]. The amplitude of PhNR and the PhNR/b-wave ratio decreases with the advancement of glaucoma, as assessed by visual field defect increases [100]. Multifocal ERG, introduced in 1992, enables numerous localized retinal responses to be identified from a single recording [101]. Research findings indicate that individuals diagnosed with glaucoma or with suspected glaucoma may exhibit diminished amplitude and/or delayed implicit time in their mfERG responses, particularly in regions that align with visual field impairments [102,103]. Several investigations have employed mfERG to differentiate between glaucoma and healthy eyes. However, thus far, none of this research has provided conclusive evidence about the diagnostic utility of the mfERG approach in glaucoma [101,104,105].

In addition, Standard Automated Perimetry (SAP) is widely regarded as a gold standard in the diagnosis, monitoring, and management of glaucoma. The field of perimetry progressed in the 1930s, when Goldmann introduced standardized kinetic perimetry. The stimuli were projected onto a hemispheric bowl and systematically moved from areas where they were not perceived to locations where they were initially detected. However, kinetic perimetry is subjective, relying on training and experience. There are significant technological restrictions, including psychophysical sensitivity to moving targets, spatial summation effects, and subject response time to the examiner. SAP, an example of static perimetry, uses a contrast stimulus. RGCs increase their firing rate when the stimulus contrast increases, until the observer detects this and responds [106]. SAP is not selective for a specific RGC type; there are newer tests that demonstrate and target specific RGCs, such as short-wavelength automated perimetry (SWAP) and frequency doubling technology (FDT). SWAP differs from SAP in its blue-light stimulus on yellow background illumination and assesses the short-wavelength cone pathway. These blue-yellow ganglion cells extend their axons towards the interlaminar, koniocellular layers of the LGN [107,108]. Researchers suggest SWAP can detect functional damage in glaucoma earlier than SAP [109,110]. Ocular hypertension patients at higher risk for glaucoma have more SWAP abnormalities than those at lesser risk [111]. However, SWAP tests take longer, which reduces patient satisfaction and healthcare practice efficiency. It may take 15% longer duration for a full-threshold SWAP than a full-threshold SAP. The introduction of the SITA threshold technique has overcome this limitation and significantly reduced test length. However, SWAP can be affected by media opacities and has greater intra-test and inter-test variability [112]. FDT measures contrast sensitivity for frequency-doubling stimulus, which is assumed to be mediated by a subpopulation of magnocellular RGCs. Several studies have demonstrated that FDT has high sensitivity and specificity in distinguishing glaucomatous patients from normal subjects, and test outcomes can be used to predict location and future onset of functional loss in suspected glaucoma as determined by SAP [113,114]. When compared to SAP and SWAP, FDT has a shorter testing time and lower variability, which may be attributed to the smaller influence of media opacities, pupil diameter, and refractive errors during testing [115]. 

A further advanced imaging modality utilized in the field of glaucoma is the Detection of Apoptosing Retinal Cells (DARC), now undergoing phase two clinical trials [116]. The method involves the intravenous administration of a fluorescent marker called Annexin V, which exhibits a strong binding affinity for phosphatidylserine. This molecule is expressed on the cell surface in apoptosis. The marker is taken up by retinal cells, which can be observed on cSLO, enabling the identification of retinal ganglion cells that are undergoing apoptosis [116]. In vivo studies have shown compatible DARC labeled cells correlated with RGC apoptosis [117,118].

Table 2 delineates the structural and functional imaging modalities employed for the assessment of RGCs in glaucoma. The refinement and integration of these techniques are indispensable for improving the diagnostic accuracy and elucidating the complex pathophysiology of RGCs in glaucomatous conditions, thereby molding the therapeutic intervention and management strategies.

## 5. Therapeutic Approaches Targeting Retinal Ganglion Cells

Therapeutic efforts concentrating on RGCs are essential for addressing the multifaceted nature of glaucoma, with the goal of slowing the disease’s progression by emphasizing cellular protection, restoration, and resilience. Although IOP reduction is the only treatment currently proven to be clinically effective in managing glaucoma, it is considered indirectly neuroprotective. Nonetheless, the development of direct neuroprotective therapies targets the neurons, specifically the RGCs themselves. A range of therapeutic approaches designed to preserve and restore RGCs have been explored within the research context for glaucoma treatment (Table 3).

### 5.1. Neuroprotective Agents

Neuroprotective agents such as brimonidine and memantine have demonstrated promise in preventing RGC degeneration by modulating neurotransmitter activity, reducing excitotoxicity, and mitigating oxidative stress. The excitotoxicity theory mentioned previously contributes to RGC degeneration through excessive glutamate release and overactivation of the NMDA receptor, and many studies have suggested the possibility of blocking the glutamate receptor as a method for RGC neuroprotection. Alpha-adrenergic receptors have been identified within RGCs, and immunohistochemical analyses demonstrated the presence of alpha-adrenergic receptors in human, bovine, and porcine retinas as well as the inner nuclear layer of the rat retina [161,162]. Brimonidine, an alpha-2 adrenergic receptor agonist, can increase retinal metabolism and neuronal growth in cultured retinal cells as well as diminish extracellular glutamate and NMDA receptor blockage [119,120,121]. A study found that continuous subcutaneous brimonidine treatment increased RGC survival under high IOP pressure for 8 weeks [122]. In addition, the Low-Pressure Glaucoma Treatment Study (LoGTS), which randomized patients to either topical brimonidine or timolol, showed 9% visual field progression in the brimonidine group and 39% progression in the timolol group [123]. However, this study had significantly greater numbers of drop-outs in the brimonidine group. In another study, brimonidine 0.2% caused less RNFL loss than timolol 0.5% in ocular hypertension patients during a 12-month period [124], which suggests the neuroprotective effect of brimonidine. MK801 (dizocilpine maleate), a strong glutamate inhibitor, is an uncompetitive NMDA antagonist with neuroprotective effects for RGC; nevertheless, due to its lengthy half-life, it can be neurotoxic [125,126]. Memantine, on the other hand, selectively and non-competitively inhibits the NMDA receptor with moderate affinity. In preclinical and experimental studies, memantine showed a partial protective effect of RGCs against glutamate toxicity when administered with low-dose glutamate in rats and enhanced RGC survival with high IOP in monkeys [127,128]. However, a Phase 3 clinical trial did not find any significant benefits compared with the placebo [129]. 

### 5.2. Calcium Channel Blockers

Calcium channel blockers (CCBs) have been examined for their neuroprotective effects against RGCs. Calcium influx into neural cells can cause apoptosis and cell death when NMDA is activated; hence, CCB has been studied for neuroprotection in glaucoma. Topical 2% flunarizine, a calcium channel blocker, reduced rabbit retinal RGC injury under high IOP ischemia conditions [130]. Brovincamine and nilvadipine have high blood–brain barrier permeability and have been shown to improve visual field progression and increase posterior choroidal circulation [131,132]. Although CCB appears promising, additional research is needed to determine its therapeutic effectiveness, as it may disrupt the autoregulation of blood flow during an acute spike in IOP [163]. 

### 5.3. Antioxidants

Oxidative stress with the increase of ROS can be detrimental to RGCs. Coenzyme Q10 (CoQ10), an electron transport chain cofactor, stabilizes mitochondrial membrane potential, supports adenosine triphosphate (ATP) synthesis, and inhibits ROS production to protect neuronal cells from oxidative stress [133,134]. CoQ10 has been shown to protect RGCs during hydrogen peroxide oxidative stress in vitro and NMDA-induced glutamate excitotoxicity in vivo [135]. It can also promote RGC survival by 29% in mouse models under oxidative stress [136]. A study found that topical Coqun eye drops (CoQ10 and vitamin E) in glaucoma patients demonstrated improved inner retinal function, indicated by PERG and visual cortex responses as determined by visual evoked potential (VEPs) [137]. 

### 5.4. Neurotrophic Factors

Beyond pharmacologic strategies, neurotrophic factor supplementation has attracted considerable interest, with a focus on the potential of BDNF, CNTF, NGF, and GDNF to promote RGC survival and resilience. These factors have demonstrated efficacy in enhancing RGC survival, reducing apoptosis, and promoting cellular resilience, highlighting their therapeutic potential in the treatment of glaucoma. Studies have shown that intravitreal BDNF injection protects RGCs in rat and primate optic nerve injury models [138,139]. In chronic intraocular hypertension, recombinant human BDNF eye drops restored PERG and VEP damage [140]. Intraocular injections of BDNF at 1.0 g/L in moderately chronic hypertensive rat eyes increased RGC survival by 2 weeks [141]; however, high-dose BDNF may significant downregulate TrkB expression, therefore lowering BDNF efficiency [142]. CNTF is particularly expressed in Muller cells, and its neuroprotective effect is mediated by glial cells, which respond by producing neurotrophic factors such as basic fibroblast growth factor [143,144]. A single vitreous injection of CNTF protein protected RGCs from nitric oxide (NO)-induced cell death and optic nerve axotomy in rats [145,146]. 

Prior research has indicated a correlation between RGC depletion and the decreased expression of NGF and its receptor in the retina. Ocular administration of NGF notably mitigated the impairments instigated by glaucoma [147]. Lambiase et al. demonstrated that NGF eye drops can mitigate glaucoma-associated optic nerve damage in rats, with higher RGC survival in treated groups with the administration of NGF eye drops four times a day over a span of 7 weeks [148]. In a separate investigation, recombinant human nerve growth factor (rh-NGF) was employed in a rat model. When rh-NGF was applied topically twice a day for 3 weeks, there was a marked enhancement in RGC survival, evidenced by diminished RGC apoptosis and an increased RGC count in the inferior retina. Additionally, topical rh-NGF was found to bolster axonal survival and curtail astrocyte activity within the optic nerve [149].

In Müller glia, GDNF intriguingly augments GLAST gene expression, which is crucial for RGC safeguarding. In mice with GLAST and EAAC1 knockouts, RGC mortality and glaucomatous injury emerged, even without elevated ocular pressure [164]. Employing this strategy, post-microsphere injections laden with GDNF, RGC longevity in DBA/2J mice was extended for nine months [150]. GDNF/Vitamin E Poly(lactide-co-glycolide) (PLGA) microsphere injections in a glaucoma animal model notably improved RGC longevity over 11 weeks, surpassing GDNF, Vitamin E, and mere microspheres [151]. Ward et al. established that GDNF via microspheres notably amplified RGC persistence in a spontaneous glaucoma model, showcasing a 3.5-fold RGC density increment in 15-month survival compared to non-treated counterparts [152]. BDNF-mediated, CNTF-mediated, NGF-mediated, and GDNF-mediated neuroprotection in glaucoma has great potential, but retinal delivery is challenging, and adverse effects must be addressed.

### 5.5. Gene Therapy

The retina presents an optimal site for gene interventions due to its accessibility, distinct functional indicators, semi-immune protection, and localized non-systemic position [165]. Recent studies suggest the potential of gene therapy in preserving both the architecture and activity of RGC. 

The γ-Synuclein (mSncg) promoter is a notable target and is of great relevance to RGCs. Through a combination of adeno-associated virus (AAV)-mSncg promoter and clustered, regularly interspaced short palindromic repeats (CRISPR)/Cas9 gene editing, pro-degenerative genes can be knocked down. Wang et al., using an AAV2-mSncg in human pluripotent stem cell (hPSC)-derived RGCs and a mice optic nerve crush model, effectively preserved the acutely injured RGC somata and axons [153].

Complement C3 is another target in gene therapy focused on RGC preservation. By overexpressing the C3 inhibitor, the activation of complement C3d is reduced. Research by Bosco et al. [154] combined AAV retinal gene therapy with the targeted C3 inhibitor through intravitreal injections in a mice glaucoma model. The result revealed neuroprotection of RGC axons and somata despite continued intraocular pressure elevation. 

In relation to RGC health, CaMKII (Calcium–calmodulin ((CaM))-dependent protein kinase II) assumes a crucial function. Augmenting the expression of CaMKII provides protection to both RGCs and their associated axons. This protective mechanism is realized through an AAV-mediated treatment involving CaMKIIa Threonine 286D, administered via intravitreal injection in a mouse glaucoma model [155]. Subsequent findings indicate that the reactivation of CaMKII safeguards the extensive axonal projections of RGCs in vivo, maintaining both visual functionalities, spanning from the retina to the visual cortex, and the integrity of visually driven behavior.

NMNAT (nicotinamide mononucleotide adenylyl transferase) represents another gene target critically linked to the well-being of RGCs. As illustrated by Fang et al., the overexpression of the long half-life NMNAT2 mutant, steered by the RGC-specific mSncg promoter in mice, facilitated by AAV2 intravitreal injection, reinstates the diminished nicotinamide adenine dinucleotide (NAD^+^) levels in RGCs and ONs affected by glaucoma. This RGC-centric gene therapy approach confers notable neuroprotective benefits to the RGC soma and its axon, while ensuring the maintenance of visual functionality in the context of the optic nerve crush model [156].

The X-linked inhibitor of apoptosis (XIAP) presents an alternative pathway for safeguarding RGCs. Through inhibition of apoptotic activation, RGCs gain both structural and functional defense. This method, implemented through intravitreal injection in a murine glaucoma paradigm, was investigated by Visuvanathan et al. [157]. Subsequent findings from PERGs demonstrated that XIAP treatment substantially preserved the function of somal and axonal RGCs in glaucomatous eyes.

The *BCLX_L_* gene emerges as a potential therapeutic target for RGCs. The activation of the BAX protein is pivotal in the intrinsic apoptotic pathway leading to RGC demise in glaucoma, with *BCLX_L_* serving as its primary counteragent. Donahue et al. engineered a mCherry-*BCLX_L_* fusion protein, inhibiting BAX localization and activation in mitochondrial tissue culture cells. Upon packaging into AAV2 and administering intravitreally in a murine glaucoma model, there was no alteration in the IOP trajectory compared to control mice. However, notable reductions were observed in RGC soma pathology and axonal deterioration in the optic nerve [158].

### 5.6. Stem Cell Therapy

Emerging evidence suggests the relevance of stem cell therapy for RGC regeneration. The potential of therapy derived from stem cells resides in its capacity to replace damaged RGCs and restore retinal functionality; however, this would require the cells to differentiate into RGCs and become integrated and capable of synapsing in precise brain locations. Stem cell transplantation from brain-derived, hippocampal-derived, and bone marrow–derived cells mostly differentiates into amacrine and horizontal cells [159]. However, a study found that Muller glia can be differentiated into RGCs, and they were able to restore the connectivity to other neurons and the negative scotopic threshold response [160]. The road to clinical application is fraught with difficulties, including assuring transplanted cells’ long-term safety and stability, optimizing delivery systems, and achieving functional integration with existing brain networks.

### 5.7. Intraocular Pressure Management

Management of IOP remains the foundational strategy in the treatment of glaucoma, with medical therapy typically serving as the initial approach to decrease IOP. Pharmacological agents, often in the form of topical eye drops, are the first line of defense to mitigate the mechanical stress exerted on RGCs and avert further deterioration and loss. Should medication not suffice, laser procedures are considered the subsequent step in the therapeutic cascade. If these measures prove inadequate, surgical interventions such as trabeculectomy or minimally invasive glaucoma surgeries (MIGS) are employed to further lower IOP. In cases where conventional surgeries do not achieve the desired outcomes, the implantation of shunts or valves may be warranted as a final recourse to manage the disease effectively [166].

### 5.8. Blood Flow Enhancement

Regeneration of the optic nerve may be facilitated by enhancing blood flow and providing robust vascular support. Optimal blood circulation is crucial for delivering nutrients and oxygen, which are essential for the survival and repair of neurons. When the optic nerve is damaged, improving blood flow can mitigate nerve degeneration and facilitate the transport of neurotrophic factors and other regenerative molecules to the site of injury. Vascular support, particularly through the stabilization of blood vessels and the prevention of ischemic damage, can further potentiate the regeneration of optic nerve fibers. Therapeutic approaches, though still largely investigational, are aimed at augmenting these physiological processes to aid in repair and regeneration. For example, ginkgo biloba extract (GBE) has been explored for its potential to improve ocular blood flow [167]. EGb 761, a refined extract of ginkgo biloba, is characterized by a reduction in ginkgolic acids, which are considered undesirable, while retaining significant concentrations of bioactive constituents, specifically flavonoid glycosides (comprising 22.0–27.0%) and terpene lactones (encompassing 5.0–7.0%) [168]. These flavonoid compounds, known for their antioxidant properties, play a role in maintaining vascular integrity, potentially offering advantages for the perfusion of the optic nerve [169]. Research involving a murine model with sustained moderate IOP elevation revealed that, following a 5-month regimen of EGb 761 administration, there was no observable impact on IOP; however, there was a notable attenuation in the degeneration of RGCs [168]. Complementary interventions, including lifestyle modifications that encourage exercise and a balanced diet, support overall vascular integrity. While these modalities offer hope, their integration into clinical practice necessitates further validation through rigorous research to establish efficacy and safety.

In conclusion, the combination of diverse therapeutic strategies, including pharmacological, neurotrophic, stem cell, and gene therapy approaches, broadens the spectrum of interventions available to clinicians, thereby facilitating the development of individualized treatment plans based on the needs and disease characteristics of each patient. These evolving therapies highlight the need for a multidimensional approach incorporating diverse therapeutic modalities to address the complexities of RGC vulnerability and pave the way for glaucoma management on a global scale. 

## 6. Conclusions and Future Directions

The pursuit of knowledge about RGCs’ role in glaucoma continues to uncover new pathways for research and innovation, necessitating a forward-thinking strategy in both the research and therapeutic arenas. Gaining a comprehensive understanding of the underlying theories pertaining to the pathophysiology of glaucoma elucidates the intricate nature of glaucomatous degeneration and holds significant relevance in identifying potential therapeutic targets for future interventions. The assessment of RGC functionality by several imaging modalities is of utmost importance in the clinical management of glaucoma. The growing interest in the cellular and molecular mechanisms behind RGC degeneration supports the development of novel therapeutic targets and therapies, which may have implications for personalized medicine methods in the management of glaucoma.

Expanding on this, the future of glaucoma research could pivot towards pioneering treatments that accomplish more than halting the progression of the disease—aiming instead to reverse its course. The exploration of neuroregenerative therapies, such as cell-based therapies, offers the tantalizing prospect of not only preserving but also restoring lost vision [170]. Moreover, advancements in nanotechnology hold the potential for targeted drug delivery systems that could administer therapeutic agents directly to RGCs, enhancing efficacy and reducing systemic side effects [171]. Artificial intelligence and machine learning algorithms could also revolutionize early detection and personalized treatment plans based on predictive modeling [172]. As we stand on the cusp of these scientific horizons, it is paramount that our research strategies incorporate these emerging modalities, ensuring a robust and dynamic approach to overcoming the challenges posed by glaucoma.

## Figures and Tables

**Figure 1 cells-12-02797-f001:**
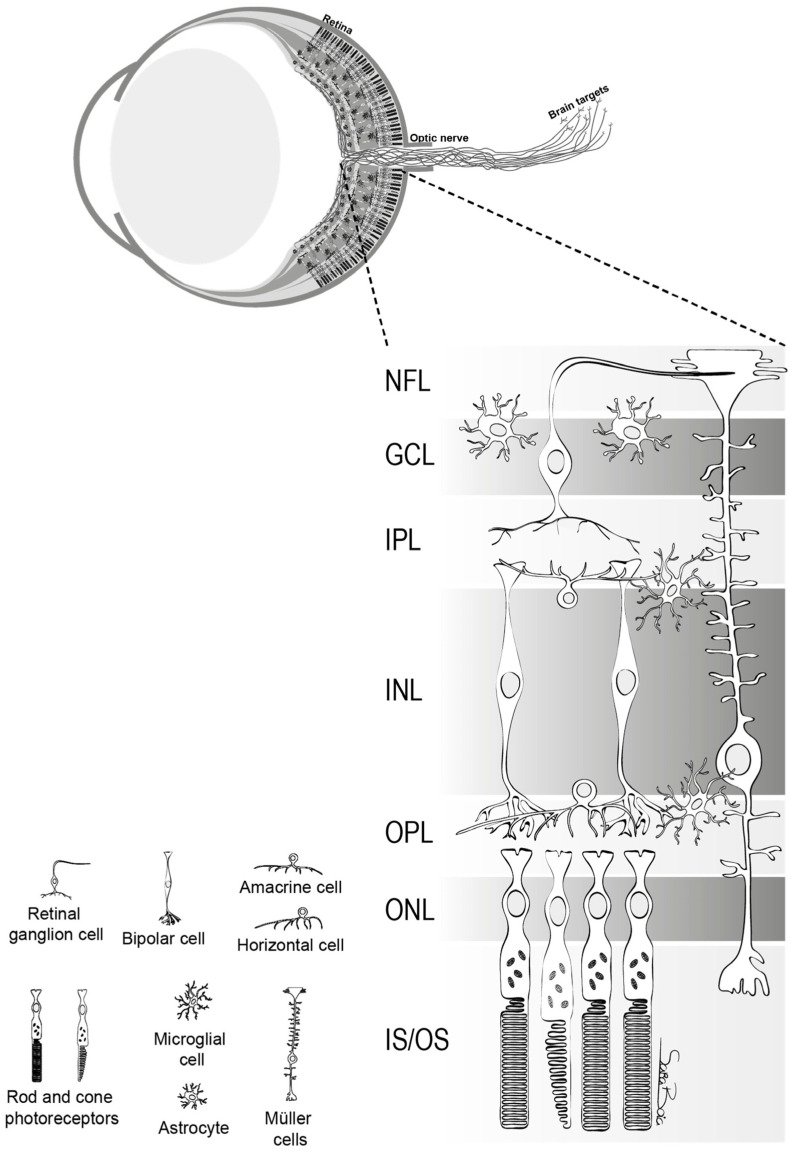
The structure of retina. This figure delineates the stratified architecture of the neural sensory retina, illustrating the cellular arrangement using distinct nuclear and plexiform layers. Photoreceptor nuclei, specifically those of rods and cones, reside within the outer nuclear layer (ONL), while the nuclei of various interneurons, including amacrine, bipolar, and horizontal cells, are primarily situated in the inner nuclear layer (INL). Retinal ganglion cells (RGCs) inhabit the ganglion cell layer (GCL), with their axonal projections extending through the nerve fiber layer (NFL). The retina comprises two macroglial variants: Müller cells, traversing the entire retinal thickness, and astrocytes, localized within the GCL. Microglia are chiefly found in the inner retinal regions and the outer plexiform layer (OPL) [5].

**Table 1 cells-12-02797-t001:** Overview of glaucoma pathophysiology and contemporary theoretical frameworks.

Theories	Key Points
Mechanical theory	- Mechanical strain, elevated IOP ^1^, and stiffness of sclera influence deformation of the lamina cribrosa (the most susceptible region in a pressurized eye). - The sclera and lamina cribrosa act as load-bearing tissues of the optic nerve head.
Vascular theory	- Vascular insufficiency can cause RGC ^2^ dysfunction and death. - Regulation for ocular blood flow is affected by OPP ^3^, temperature, and neural function. - Vascular autoregulation and neurovascular coupling can be compromised in glaucoma. - Mediators such as endothelin-1 and nitric oxide can affect blood flow and RGC health.
Excitotoxicity theory	- Excitotoxicity involves excessive glutamate release, leading to overstimulation of NMDA ^4^ receptors on RGCs, resulting in calcium influx, oxidative stress, mitochondrial dysfunction, and RGC injury.
Neurotrophic factor deprivation theory	- Interruption of axonal transport leads to reduced neurotrophic factors essential for RGC survival. - BDNF ^5^, produced in the retina, is mediated through the TrkB ^6^ receptor, and its activation has a significant impact on RGC survival. Elevated IOP may hinder BDNF transport, resulting in RGC apoptosis and loss. - Other neurotrophic factors linked to RGC health: CNTF ^7^, NGF ^8^, and GDNF ^9^.
Oxidative Stress and Inflammation	- Elevated IOP, ischemia-reperfusion injury, enzymatic degradation of neurotransmitters, and mitochondrial dysfunction contribute to ROS ^10^ overproduction and inflammatory mediators and can exacerbate RGC death. - The trabecular meshwork is particularly vulnerable to oxidative damage. - Elevated IOP can activate innate immune responses with increased microglial activity and complement expression. - Adaptive immunity in glaucoma has been observed with induction of HSP-specific ^11^ memory T cells.

^1^ IOP, intraocular pressure; ^2^ RGC, retinal ganglion cell; ^3^ OPP, ocular perfusion pressure; ^4^ NMDA, N-methyl-d-aspartate; ^5^ BDNF, brain-derived neurotrophic factor; ^6^ TrkB, tropomyosin receptor kinase B; ^7^ CNTF, ciliary neurotrophic factor; ^8^ NGF, nerve growth factor; ^9^ GDNF, glial cell line–derived neurotrophic factor; ^10^ ROS, reactive oxygen species; ^11^ HSP, heat shock protein.

**Table 2 cells-12-02797-t002:** Imaging modalities for assessing structural and functional retinal ganglion cells in glaucoma.

Retinal Imaging	Description	Applications	Advantages	Disadvantages
Optical coherence tomography (OCT)	A non-invasive imaging technique that uses light waves to take high-resolution cross-sectional pictures of the retina.	Measurement of RNFL ^1^ thickness, optic nerve, and macular ganglion cell complex	High-resolution images and non-invasive	Affected by media opacities (e.g., cataract), poor visualization of RGCs ^2^
Confocal scanning laser ophthalmoscopy (cSLO)	cSLO utilizes a narrow laser beam to scan the retina point by point through a small aperture.	Visualization of optic nerve head	High contrast image, excellent lateral resolution, and monitoring of RNFL defects and glaucomatous changes	Affected by media opacities (e.g., cataract) and poor axial resolution
Adaptive optics imaging	Adaptive optics imaging employs deformable mirrors and wavefront sensors to correct optical aberrations.	Visualization of RGC	High-resolution, allowing visualization of individual cells, and can correct optical aberrations	Pupil dilation, longer time needed, and costly
Electroretinography (ERG)	ERG measures the electrical responses of various cell types in the retina after visual stimulation.	Assesses the functional status of RGCs using PERG ^3^, PhNR ^4^, and mfERG ^5^	Functional assessment of RGCs, objective measurement of retinal activity, may detect early glaucomatous changes	Affected by media opacities, interpretation based on test parameters, utilization still under evaluation
Standard automated perimetry (SAP)	SAP assesses the visual field by presenting light stimuli of varying intensities in a standardized pattern to detect visual deficits.	The central tool in glaucoma diagnosis, staging, and monitoring. Visual field defects correlate with RGC loss and damage to the optic nerve.	Widely accepted and utilized, objective measures of visual function, able to monitor progression, offering other specific techniques in assessing specific RGC types, such as SWAP ^6^ and FDT ^7^	Dependent on patient’s cooperation and understanding, can be affected by media opacities and ptosis, test–retest variability
Detection of apoptosing retinal cells (DARC)	DARC utilizes fluorescently tagged annexin-V intravenously to mark apoptotic RGCs, which are then identified with cSLO.	To assess efficacy of therapeutic agents	Identification of individual apoptosis RGC cells and early detection of glaucomatous damage	Invasive, long-term effects not yet known

^1^ RNFL, retinal nerve fiber layer; ^2^ RGC, retinal ganglion cell; ^3^ PERG, pattern reversal ERG; ^4^ PhNR, photopic negative response; ^5^ mfERG, multifocal ERG; ^6^ SWAP, short-wavelength automated perimetry; ^7^ FDT, frequency doubling technology.

**Table 3 cells-12-02797-t003:** Therapeutic targets for retinal ganglion cells.

Category	Agent/Method	Effect/Outcome	Reference
Neuroprotective agents	Brimonidine	Alpha-2 adrenergic receptor agonist; reduces extracellular glutamate, blocks NMDA ^1^ receptor, and supports RGC ^2^ survival under high IOP ^3^	[119,120,121,122,123,124]
	MK801 (dizocilpine maleate)	Strong glutamate inhibitor; uncompetitive NMDA antagonist with neuroprotective effects; can be neurotoxic due to long half-life	[125,126]
	Memantine	Selective and non-competitive NMDA receptor inhibitor; showed partial RGC protection in preclinical studies but no significant benefit in Phase 3 clinical trial	[127,128,129]
Calcium channel blockers	Topical 2% flunarizine	Reduced RGC injury in rabbits under high IOP conditions	[130]
	Brovincamine and nilvadipine	Showed improved visual field progression and increased posterior choroidal circulation	[131,132]
Anti-oxidative agents	Coenzyme Q10 (CoQ10)	Protects RGCs from oxidative stress, supports ATP ^4^ synthesis, inhibits ROS production; promotes RGC survival in oxidative stress models	[45,133,134,135,136,137]
Neurotrophic factors	Brain-derived neurotrophic factor (BDNF)	Intravitreal injection protects RGCs in animal models; restores PERG and VEP damage; increases RGC survival in hypertensive rat eyes with intraocular injections	[138,139,140,141,142]
	Ciliary neurotrophic factor (CNTF)	Expressed in Muller cells; offers protection against NO-induced ^5^ cell death and optic nerve axotomy in rats	[143,144,145,146]
	Nerve growth factor (NGF)	NGF eye drops can mitigate glaucoma-associated optic nerve damage in rats, with higher RGC survival; topical rh-NGF ^6^ results in improved RGC survival, decreased apoptosis, and reduced astrocyte activity in the optic nerve in a rat model	[147,148,149]
	Glial cell line–derived neurotrophic factor (GDNF)	Enhances GLAST gene expression in Müller glia, essential for RGC protection; GDNF microspheres increase RGC survival and density in rat models	[150,151,152]
Gene therapy	γ-synuclein (mSncg) promoter	Preserves the acutely injured RGC somata and axons	[153]
	Complement C3	Neuroprotection of RGC axons and somata	[154]
	Calcium/calmodulin-stimulated protein kinase II (CaMKII)	Protection of RGCs and their axons	[155]
	Nicotinamide mononucleotide adenylyl transferase (NMNAT)	Significant neuroprotection of both RGC soma and axon and preservation of visual function	[156]
	X-linked inhibitor of apoptosis (XIAP)	Provide both functional and structural protection of RGC	[157]
	*BCLX_L_*	Robustly attenuate both RGC soma pathology and axonal degeneration in the optic nerve	[158]
Stem cell Therapy	Brain-derived, hippocampal-derived stem cells	Differentiate into amacrine and horizontal cells	[159]
	Muller glia stem cells	Differentiate into RGCs, restore connectivity to other neurons, and address negative scotopic threshold response	[160]
IOP management	Trabeculectomy and Pharmacological Medications	Essential for reducing mechanical stress on RGCs, preventing further degeneration	

^1^ NMDA, N-methyl-d-aspartate; ^2^ RGC, retinal ganglion cell; ^3^ IOP, intraocular pressure; ^4^ ATP, adenosine triphosphate; ^5^ NO, nitric oxide; ^6^ rh-NGF, recombinant human nerve growth factor.

## Data Availability

Not applicable.

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
