# Peer review of "The Role of Retinal Ganglion Cell Structure and Function in Glaucoma"

_cells, 2023, doi:10.3390/cells12242797_

Round 1

Reviewer 1 Report

Comments and Suggestions for Authors

This is a great review paper to discuss the role of retinal ganglion cell in glaucoma filed. Manuscript is showing well organized and propriate construction to help audience get to know the whole picture of RGC according to current scientific research.

I may have some suggestions:

1. in the Part 2, basic anatomy and function of RGC:

Regarding to discussion of anatomy and morphology of various of subtypes of RGC, if the author can attach some pictures showing what they look like. I believe picture will make subtypes more impressive and understandable. For example, normal morphology, and injured cell which shows soma shrinkage.

2. in the Part 3, pathophysiology of glaucoma, may need to add "mitochondria dysfunction" content into stress and inflammation theory, which is hot topic recently should not be ignored.

3.in the Part 5 therapeutic approaches, please add "optic nerve regeneration" "enhancing blood flow and vascular support" related content.

Comments on the Quality of English Language

Lind 67, "is" should be lower case. Looks like a typo.

Author Response

Response to Reviewer 1 Comments

Point 1: In the Part 2, basic anatomy and function of RGC: Regarding to discussion of anatomy and morphology of various of subtypes of RGC, if the author can attach some pictures showing what they look like. I believe picture will make subtypes more impressive and understandable. For example, normal morphology, and injured cell which shows soma shrinkage.

Response 1: We express our gratitude for the meticulous review and the insightful feedback provided. In adherence to your recommendations, we have amended the pertinent sections and incorporated Figure 1 to enhance the clarity and comprehensibility of the content. Please see the attached manuscript for the revised paragraphs.

Point 2: In the Part 3, pathophysiology of glaucoma, may need to add "mitochondria dysfunction" content into stress and inflammation theory, which is hot topic recently should not be ignored.

Response 2: Thank you for your valuable comments. The concept of “mitochondria dysfunction” is emphasized and added to the paragraphs in Part 3. Please see the attached manuscript for the revised paragraphs.

Point 3: In the Part 5, therapeutic approaches, please add "optic nerve regeneration" "enhancing blood flow and vascular support" related content.

Response 3: We appreciate your insightful comments and the guidance provided. Based on your suggestion, the potential therapies regarding enhancing blood flow and vascular support for optic nerve regeneration are added in the Part 5.8. Please see the attached manuscript for the revised paragraphs.

Comments on the Quality of English Language: Lind 67, "is" should be lower case. Looks like a typo.

Response 4: Thanks for your careful review. We have corrected the typo as you mentioned.

Reviewer 2 Report

Comments and Suggestions for Authors

This a very nicely designed and put together review paper about a major optic neuropathy affecting a large number of people. 

It is a very comprehensive review covering virtually all aspect of glaucoma types of diseases. A minor concern is that future directions of research section is a bit brief and could have been extended to include potential and even  futuristic modalities of treatments that can reverse the course of the disease.

Author Response

Response to Reviewer 2 Comments

Point: A minor concern is that future directions of research section is a bit brief and could have been extended to include potential and even futuristic modalities of treatments that can reverse the course of the disease.

Response: We appreciate your insightful comments and the guidance provided. In response to your recommendations, we have revised the relevant paragraphs, offering more comprehensive future directions and treatments under study. The revised content is enclosed in the attached manuscript.

Reviewer 3 Report

Comments and Suggestions for Authors

This manuscript by Kathy Ming Feng et al. presents a review of the role of retinal ganglion cell (RGC) structure and function in glaucoma.  The narrative examines the anatomy of RGC subtypes, and covers the various underlying theoretical mechanisms that lead to RGC susceptibility in glaucoma (mechanical, vascular, excitotoxicity, neurotrophic factor deficiency, and oxidative stress and inflammation).  Imaging methodologies and functional assessments of RGCs are also discussed.  There is also some discussion regarding potential for neuroprotective therapy to manage and treat glaucoma in the future.

In general, the manuscript is well-written and will prove to be a useful resource for researchers and clinicians in the field.

I have some minor comments, as outlined below:

 Basic anatomy and function of retinal ganglion cells (lines 53-89). 

This section would be much enhanced with some figures illustrating the morphology of the major different RGC types, rather than just verbal descriptions.  Also, please state how many different types of RGCs are known to exist in the human retina, perhaps some other species commonly used in glaucoma research as well.

Line 67, typographical error: ‘Is a fundamental quality’ should be written as ‘is a fundamental quality’.  

Lines 69-70, typographical error: ‘account’ should be written as ‘accounting’. 

Line 86, typographical error: ‘in pressure-induced environment’ should be written as ‘in a pressure-induced environment’.

Table 1.  First line of Key Points associated with Mechanical theory, ‘elevated IOP1’.  Should this not read as ‘elevated IOP1’?  Also, Vascular Theory.  ‘endothelium-1’ should be written as ‘endothelin-1’.  Similarly, on line 163, under the Vascular Theories sub-header, ‘endothelin-1’, not ‘endothelium-1’.

In the endothelin-1 discussion, only 3 references are cited, which are quite dated.  Please include some more recent citations.

Neurotrophic factor deprivation theory, line 235.  Close parentheses.  Should be written as ‘phospholipase C-gamma (PLC-gamma))’, not ‘phospholipase C-gamma (PLC-gamma)’.

Line 274, typographical error: ‘Normally the eye is regarded’, not ‘Normally eye is regarded’

Line 286.  Does the word ‘It’ here refer to microglia?  Please clarify.

Imaging techniques for assessing retinal ganglion cells.  Line 328 onwards.  This section is not describing imaging techniques for RGCs but functional evaluation of RGCs.  It needs a separate sub-header.

Line 338.  Do these changes in PERG and RNFL (presumably the authors mean a reduction in PERG amplitude and thinning of the RNFL) take place in a glaucomatous eye?

Line 354.  Please change ‘Goldman’ to 'Goldmann’.  This is after Hans Goldmann (1899-1991).

Therapeutic approaches targeting retinal ganglion cells, line 401.  Please add a sentence here indicating clearly that, to date, the only clinically efficacious treatment for glaucoma is reduction of intraocular pressure, despite experimental advances in neuroprotection.

Typographical error, line 427.  ‘greater number of drop-outs in the brimonidine group’, not ‘greater number of dropped out in the brimonidine group’. 

Typographical error, line 547.  Replace word ‘synapse’ with ‘synapsing’

Line 556.  The word ‘Moreover’ is redundant here.  Also, this entire short section, Intraocular pressure management, would read better if it emphasized that medical therapy is the first line of treatment to lowers IOP, followed by laser procedures, then trabeculectomy and/or minimally invasive glaucoma surgery (MIGS), and finally implantation of shunts or valves.    

Comments on the Quality of English Language

Very minor editing suggested, as outlined above.

Author Response

Response to Reviewer 3 Comments

Basic anatomy and function of retinal ganglion cells (lines 53-89).

This section would be much enhanced with some figures illustrating the morphology of the major different RGC types, rather than just verbal descriptions. Also, please state how many different types of RGCs are known to exist in the human retina, perhaps some other species commonly used in glaucoma research as well.

Response: We express our gratitude for the meticulous review and the insightful feedback provided. In adherence to your recommendations, we have amended the pertinent sections and incorporated Figure 1 to enhance the clarity and comprehensibility of the content. Please see the attached manuscript for the revised paragraphs.

Line 67, typographical error: ‘Is a fundamental quality’ should be written as ‘is a fundamental quality’. 

Lines 69-70, typographical error: ‘account’ should be written as ‘accounting’.

Line 86, typographical error: ‘in pressure-induced environment’ should be written as ‘in a pressure-induced environment’.

Response: Thanks for your valuable careful review. We have revised the typographical errors.

Table 1.  First line of Key Points associated with Mechanical theory, ‘elevated IOP1’.  Should this not read as ‘elevated IOP1’?  Also, Vascular Theory.  ‘endothelium-1’ should be written as ‘endothelin-1’.  Similarly, on line 163, under the Vascular Theories sub-header, ‘endothelin-1’, not ‘endothelium-1’.

Response: Thanks for your valuable careful review. We have revised the typographical errors.

In the endothelin-1 discussion, only 3 references are cited, which are quite dated. Please include some more recent citations.

Response: We express our gratitude for the meticulous review and the insightful feedback provided. We have added more recent references according to your advice and rewrited the paragraph.

Neurotrophic factor deprivation theory, line 235.  Close parentheses.  Should be written as ‘phospholipase C-gamma (PLC-gamma))’, not ‘phospholipase C-gamma (PLC-gamma)’.

Line 274, typographical error: ‘Normally the eye is regarded’, not ‘Normally eye is regarded’

Response: Thanks for your valuable careful review. We have revised the typographical errors.

Line 286.  Does the word ‘It’ here refer to microglia?  Please clarify.

Response: Thank you for pointing out the need for clarification in Line 286 of our manuscript. We have now revised the sentence to eliminate any ambiguity regarding this reference.

Imaging techniques for assessing retinal ganglion cells. Line 328 onwards. This section is not describing imaging techniques for RGCs but functional evaluation of RGCs.  It needs a separate sub-header.

Response: We express our gratitude for the meticulous review and the insightful feedback provided. In adherence to your recommendations, we have adjusted the paragraphs with separate subheaders. Please see the attached manuscript for the revised paragraphs.

Line 338.  Do these changes in PERG and RNFL (presumably the authors mean a reduction in PERG amplitude and thinning of the RNFL) take place in a glaucomatous eye?

Response: We appreciate your insightful comments and the guidance provided. Yes, the changes in pattern electroretinography (PERG) and retinal nerve fiber layer (RNFL) typically manifest as a reduction in PERG amplitude and thinning of the RNFL in a glaucomatous eye. The evidence suggests a strong correlation between the presence of glaucomatous damage and these specific alterations. Therefore, we have revised the paragraph to reflect your valuable suggestion, clarifying this relationship.

Line 354.  Please change ‘Goldman’ to 'Goldmann’.  This is after Hans Goldmann (1899-1991).

Response: Thanks for your valuable careful review. We have revised the typographical errors.

Therapeutic approaches targeting retinal ganglion cells, line 401.  Please add a sentence here indicating clearly that, to date, the only clinically efficacious treatment for glaucoma is reduction of intraocular pressure, despite experimental advances in neuroprotection.

Response: We appreciate your insightful comments and the guidance provided. The concept is added to the paragraph.

Typographical error, line 427.  ‘greater number of drop-outs in the brimonidine group’, not ‘greater number of dropped out in the brimonidine group’.

Typographical error, line 547.  Replace word ‘synapse’ with ‘synapsing’

Response: Thanks for your valuable careful review. We have revised the typographical errors.

Line 556.  The word ‘Moreover’ is redundant here.  Also, this entire short section, Intraocular pressure management, would read better if it emphasized that medical therapy is the first line of treatment to lowers IOP, followed by laser procedures, then trabeculectomy and/or minimally invasive glaucoma surgery (MIGS), and finally implantation of shunts or valves.   

Response: We appreciate your insightful comments and the guidance provided. In response to your recommendations, we have revised the relevant paragraphs and the revised content is enclosed in the attached manuscript.